# The nociceptive response induced by different classes of *Tityus serrulatus* neurotoxins: The important role of Ts5 in venom-induced nociception

Felipe Cerni[1☯], Isadora Oliveira[2☯], Francielle Cordeiro[2], Karla Bordon[2], Isabela Ferreira[2], Wuelton Monteiro[3,4], Eliane Arantes[2], Thiago Cunha[5], Manuela Pucca[1,6]*

1 Health and Sciences Postgraduate Program, Federal University of Roraima, Boa Vista, Roraima, Brazil, 2 Department of BioMolecular Sciences, School of Pharmaceutical Sciences of Ribeirão Preto, University of São Paulo, Ribeirão Preto, São Paulo, Brazil, 3 Department of Teaching and Research, Dr. Heitor Vieira Dourado Tropical Medicine Foundation, Manaus, Amazonas, Brazil, 4 Department of Medicine and Nursing, School of Health Sciences, Amazonas State University, Manaus, Amazonas, Brazil, 5 Center for Research in Inflammatory Diseases (CRID), Department of Pharmacology, Ribeirão Preto Medical School, University of São Paulo, Ribeirão Preto, São Paulo, Brazil, 6 Medical School, Federal University of Roraima, Boa Vista, Roraima, Brazil

☯ These authors contributed equally to this work.
* manu.pucca@ufrr.br

**Data Availability Statement:** All relevant data are within the manuscript and its Supporting information files.

## Abstract

Scorpion sting envenomations (SSE) are feared by the intense pain that they produce in victims. Pain from SSE is triggered mainly by the presence of neurotoxins in the scorpion venom that modulates voltage-gated ion channels. In Brazil, SSE is mostly caused by *Tityus serrulatus*, popularly known as yellow scorpion. Here, we evaluated experimental spontaneous nociception induced by *T. serrulatus* venom as well as its isolated neurotoxins Ts1, Ts5, Ts6, Ts8, and Ts19 frag II, evidencing different degrees of pain behavior in mice. In addition, we developed a mice-derived polyclonal antibody targeting Ts5 able to neutralize the effect of this neurotoxin, showing that Ts5 presents epitopes capable of activating the immune response, which decreased considerably the nociception produced by the whole venom. This is the pioneer study to explore nociception using different classes of *T. serrulatus* neurotoxins on nociception (α-NaTx, β-NaTx, α-KTx, and β-KTx), targeting potassium and sodium voltage-gated channels, besides demonstrating that Ts5 plays an important role in the scorpion sting induced-pain.

## Author summary

The Brazilian *Tityus serrulatus* scorpion envenoming is well recognized to be responsible for its painful sting. The pain reported by victims of *T. serrulatus* has shown variations, varying from mild to severe. However, little is known regarding the mechanisms involving the venom-derived toxins and their relationship a role in causing nociception. This research pioneer shows how each *T. serrulatus* classes of neurotoxins (Ts1: β-NaTx; Ts5:

**Funding:** We thank Fundação de Amparo à Pesquisa do Estado de São Paulo (FAPESP, São Paulo Research Foundation, scholarship to IO n. 2020/13176-3, grants to EA n. 2019/10173-6, and TC n. 2013/08216-2), Coordenação de Aperfeiçoamento de Pessoal de Nível Superior (CAPES, Coordination for the Improvement of Higher Education Personnel, scholarship to IF finance code 001), Conselho Nacional de Desenvolvimento Científico e Tecnológico (CNPq, The National Council for Scientific and Technological Development, scholarships to MP n. 307184/2020-0, WM n. 309207/2020-7, EA n. 306479/2017-6, and FCordeiro n. 155276/2018-2). WM acknowledges funding support from Fundacão de Amparo à Pesquisa do Estado do Amazonas (FAPEAM, PAPAC 005/2019, PRO-ESTADO and Posgrad calls). MP thanks the PROPESQUISA/PRPPG-UFRR. The funders had no role in study design, data collection and analysis, decision to publish, or preparation of the manuscript.

**Competing interests:** The authors have declared that no competing interests exist.

α-NaTx; Ts6: α-KTx; Ts8 and Ts19: β-KTx) are involved in the spontaneous nociception. The new knowledge may contribute to a better overall understanding of the mechanisms underlying *T. serrulatus* painful envenomings.

## Introduction

In Brazil, scorpion sting envenomations (SSE) are considered as a public health problem reaching more than 105,000 occurrences in 2021 [1]. *Tityus serrulatus* (Ts), popularly known as yellow scorpion, is the species responsible for most Brazilian SSE cases, mainly due to its adaptation to urban environments and its parthenogenic reproduction [2–4]. Moreover, it is considered the most dangerous scorpion in the country, causing considerable morbidity and lethality [3].

The signals and symptoms following Ts stings may be mild, moderate, or severe. During mild envenomings, victims report intense pain, which may come with paresthesia, nausea, and sweating. Moderate envenomings are represented by mild signs and symptoms along with bradycardia or tachycardia, hypo, or hypertension, vomiting, diarrhea, and dyspnea. The severe envenomings are characterized by cardiac failure, pulmonary edema, and death, mainly affecting children and elderly [2, 5].

Independent of the envenoming severity, intense pain is a hallmark of Ts stings, which is induced mainly by venom-derived neurotoxins [3, 6] able to modulate sodium channels (NaTxs) or potassium channels (KTxs) [7–10]. Furthermore, these neurotoxins may also release histamine, 5-hydroxytryptamine, and other pro-inflammatory mediators (*e.g.*, cytokines), amplifying the nociceptive response [11–14]. Ts venom (TsV) is composed by neurotoxins that bind specifically to voltage-gated sodium channels or Navs (*e.g.*, Ts1, Ts2, Ts4, and Ts5) [15–18] as well as neurotoxins targeting voltage-gated potassium channels or Kvs (*e.g.*, Ts6, Ts7, Ts8, Ts9, Ts15, and Ts19 fragments I and II) [7, 19–21]. There are several Navs associated to pain induction, such as the isotypes Nav1.3, Nav1.7, Nav1.8, and Nav1.9 [22]. Regarding Kvs family, Kv4.2 is the main studied channel inducing hyperalgesia [23]. Although a great number of studies with TsV and neurotoxins have been conducted so far, few have focused on nociceptive response [11, 12], with only two being conducted with isolated Ts components [7, 24].

Here, we evaluated experimental spontaneous nociception induced by TsV as well as by its isolated neurotoxins Ts1, Ts5, Ts6, Ts8, and Ts19 frag II. In addition, we developed a mice-derived polyclonal antibody targeting Ts5 in the attempt to abolish the nociception produced by this toxin in the whole venom. The approach used here besides helping to understand the mechanisms of pain in the SSE and opens the way to potential treatments to minimize the harmful effects for Ts-envenomed patients.

## Material and methods

### Ethics statement

All protocols were approved by the Institutional Committee for Animal Care and Use of the University of São Paulo (n. 97/2015/USP) and were performed following ethical guidelines established for investigations of experimental pain in conscious animals.

### *T. serrulatus* venom and toxins

TsV was extracted using electrical stimulation method (12 V) and fractionated through cation-exchange chromatography [19]. Ts fractions were rechromatographed according to previously

described methods to obtain isolated NaTx and KTx channel toxins. Briefly, Ts1 (β-NaTx) and Ts5 (α-like-NaTx) were obtained from fractions XIII and XIA [16, 25, 26], representing 16% and 2% of soluble venom [27], respectively, while Ts6 (α-KTx), Ts8 (β-KTx), and Ts19 frag II (β-KTx) were isolated from fractions X, VIIIA/VIIIB, and VIB and represent 1.82%, 0.008%, and 0.03% from soluble venom, respectively [7, 19, 21, 28]. Toxins' concentrations were determined using absorbance at 280 nm and the molar extinction coefficients [29, 30]. Details regarding classification, recovery, molecular weight, and ion channels' binding are summarized in Table 1.

### *In vivo* nociceptive assay

**Animals.** Male C57BL mice (22–26 g) were obtained from the housing facilities of the Bioterium of Ribeirão Preto Campus of University of São Paulo (USP). Animals were maintained in a temperature-controlled room (22 ± 1 ˚C) under a 12 h light/12 h dark cycle with free access to food and water. All protocols were performed following ethical guidelines established for investigations of experimental pain in conscious animals [34]. Mass, concentrations, and molarity of tested toxins are summarized in Table 2.

### Spontaneous nociception

For spontaneous nociceptive test, animals (n = 6) were challenged with intraplantar injection of TsV or its isolated toxins (Ts1, Ts5, Ts6, Ts8, and Ts19 frag II) using 0.5 μg/paw (right hind paw). Ts5 nociceptive response was also tested using different concentrations of the toxin (0.1, 0.25, 0.5 and 4 μg/paw). Control animals were challenged by the injection of saline solution (vehicle). All animals were observed for 35 min to record the time in which the animals spent either licking or lifting/shaking the injected paw. The spontaneous nociception assay was repeated with TsV after neutralizing Ts5 from the venom (see method of Ts5 neutralization below).

### Generation of polyclonal antibodies targeting Ts5

**Mice immunization.** For mice immunization, 3 Balb/c male mice (18–22 g) were challenged based to previous protocol [35], with modifications. Briefly, each animal received intramuscularly (i.m.) 7.5 μg of Ts5 antigen emulsified in a solution containing complete Freunds' adjuvant, followed by two sequential i.m. boosts with 3-week intervals of 7.5 μg and 5 μg/animal, respectively. The second boost with Ts5 was diluted in a solution with incomplete Freunds' adjuvant, and the third consist only by the antigen (Ts5). Control animals (n = 2) received solely saline solution (vehicle).

### Mice IgG isolation

For mice anti-Ts5 antibody isolation, the NAb™ Protein G Spin Kit (ref. 89949, Thermo Scientific, Waltham, MA, USA) was used, according to manufacturer's instructions. A 8% SDS-PAGE [36] was performed to evaluate IgG isolation, under reducing conditions. Molecular marker (97.0–14.4 kDa, 17-0446-01, GE Healthcare, Uppsala, Sweden) and Coomassie stain were used.

### ELISA

For monitoring immunization, a standard indirect ELISA was performed. A 96-well plate (ref. 3590, Corning, New York, EUA) was coated with Ts5 (2 μg/well, n = 2) in 0.05 M carbonate/bicarbonate buffer, pH 9.6 (100 μL/well), and incubated overnight at 4 ˚C. As positive control,

**Table 1. Molecular and electrophysiological characteristics of the neurotoxins used in the study.**

| Toxin | Access number[1] | Sequence | Class | Tested channels | Affinity | Recovery (%) | ε 280nm (0.1%)[2] | MW (Da)[3] | Ref. |
|---|---|---|---|---|---|---|---|---|---|
| Ts1 | P15226 | KEGYLMDHEGCKLSCFIRPSGYCGRECGIKKGSSGYCAWPACYCYGLPNWVKVWDRATNKCG | β-NaTx | Nav (1.1–1.8, DmNav1, NaChBac) | Nav (1.2–1.6, DmNav1) | 16 | 3.548 | 6,878 | [15] |
| Ts5 | P46115 | KKDGYPVEGDNCAFACFGYDNAYCDKLCKDKKADDGYCVWSPDCYCYGLPEHILKEPTKTSGRC | α-like-NaTx | Nav (1.2–1.8, DmNav1, NaChBac) | Nav (1.2–1.7, DmNav1) | 2 | 2.075 | 7,187 | [16] |
| Ts6 | P59936 | WCSTCLDLACGASRECYDPCFKAFGRAHGKCMNNKCRCYT | α-KTx | Kv (1.1–1.6; 2.1, 3.1, 4.3, 7.1, 7.2, 7.4, hERG, Shaker IR), | Kv (1.2, 1.3, Shaker IR) | 1.82 | 1.989 | 4,503 | [19] |
| Ts8 | P69940 | KLVALIPNDQLRSILKAVVHKVAKTQFGCPAYEGYCNDHCNDIERKDGECHGFKCKCAKD | β-KTx | Kv (Shaker IR, 1.1–1.6; 2.1, 3.1, 4.2, 7.1, 7.2, 7.4, 7.5 10.1), Nav (1.2, 1.4, 1.6, BgNav) | Kv 4.2 | 0.008 | 0.499 | 6,712 | [7] |
| Ts19 frag II | P86822 | KDKMKAGWERLTSQSEYACPAIDKFCEDHCAAKKAVGKCDDFKCN | β-KTx | Nav (1.2, 1.4–1.6, BgNav), Kv (1.1–1.6; 2.1, 3.1, 4.2, 7.1, 7.2, 7.4, 7.5 10.1, hERG, Shaker IR) | Kv 1.2 | 0.03 | 1.264* | 5,519 | [21] |
| TsV | N.A. | N.A. | N.A. | N.A. | N.A. | N.A. | 1.65** | N.A. | [31] |

[1][32];
[2][27];
[3][33];

*Theoretical extinction coefficient (280 nm) determined using the ProtParam program (www.expasy.org);

**[31]. N.A.: not applicable.

**Table 2. *T. serrulatus* venom and toxins: Mass and molarity used for nociceptive assays.**

| Toxin | 1 | 2 | 3 |
|---|---|---|---|
| | Mass at 280nm (µg) | Mass using extinction coefficient (ε) (µg) | Moles calculated using column 2 (moles) |
| Ts1 | 0.5 | 0.141 | $20.5 \times 10^{-6}$ |
| Ts5 | 0.5 | 0.241 | $33.5 \times 10^{-6}$ |
| Ts6 | 0.5 | 0.251 | $0.055 \times 10^{-3}$ |
| Ts8 | 0.5 | 1.002 | $0.15 \times 10^{-3}$ |
| Ts19 frag II | 0.5 | 0.395 | $0.07 \times 10^{-3}$ |
| TsV | 0.5 | 0.309 | n.a. |

TsV—*T. serrulatus* venom. n.a.—not applicable.

a non-immune mouse serum was used to coat the plate. The plates were washed 3 times with phosphate buffered saline (PBS) pH 7.2 for 3 times, blocked with 250 µL of PBS containing 2% (w/V) non-fat dry milk (MPBS, Molico, Nestlé, Bebey, Switzerland), and incubated for 2 h at 37 ˚C. Plates were washed again with PBS-0.05% Tween (PBS-T) and PBS, 3 times in both, and the animal's serum immunized with Ts5 (diluted 2:100 in 1% MPBS) were added in the wells and incubated for 1 h at 37 ˚C. As negative control, control animals' sera were used instead of animals' serum immunized with Ts5. The plate was washed as previously, and 100 µL of anti-mouse polyclonal antibodies conjugated with peroxidase (Anti-mouse IgG-HRP, A9044, Sigma-Aldrich, St. Louis, MO, USA, 1:1000 in 1% MPBS) for 1 h at room temperature. The plates were washed again as described above with PBS-T and PBS. After that, 100 µL of *o*-phenylenediamine dihydrochloride (OPD-$H_2O_2$, SIGMAFAST OPD tablet, Sigma-Aldrich, St. Louis, MO, USA) were added in each well and incubated for 15 min at room temperature, in dark, for color development. The reaction was stopped with 50 µL of 1 M sulfuric acid ($H_2SO_4$, Merck, São Paulo, SP, Brazil) and absorbance reading was performed at 490 nm on a microplate reader (Sunrise-basic Tecan, Männedorf, Switzerland).

After verifying the anti-Ts5 antibody development and perform the whole IgGs isolation from the immunized mice serum, the same ELISA described above assay was reproduced with purified IgGs.

## Ts5 neutralization

To neutralize the Ts5 from the whole TsV, we considered the percentage of Ts5 recovery from TsV (~2%) [27]. Based on that, we calculate anti-Ts5 IgGs 2-times the equimolar (eqM) quantity of Ts5 from the whole venom. A total of 0.5 **µg**/animal paw of TsV were used in positive control animals. In the tested group of animals, 0.5 **µg** of Ts5/animal paw (33.5 pmoles and 241 ng) was pre-incubated 1 h at 37 ˚C with 2 eqM of IgGs anti-Ts5 (67 pmoles and 10.05 **µg**). Knowing that IgGs are bivalents, hypothetically, the ratio of binding site and antigen would be 4:1 (two Ts5 binding sites for each IgG). The same rate was used in the TsV and anti-Ts5 group, using 0.5 **µg** of TsV/animal paw pre-incubated 1 h at 37 ˚C with IgGs anti-Ts5 (2 eqM of Ts5).

## Immunorecognition of Ts5 toxin by heterologous scorpion antivenom and B-cell epitope prediction

A new indirect ELISA was performed as described in 2.3.3 item to evaluate the ability of the commercial scorpion antivenom to recognize Ts5. Plates were coated with TsV and Ts5 (2 µg/

well, n = 2) in 0.05 M carbonate/bicarbonate buffer, pH 9.6 (100 μL/well), and incubated overnight at 4 ˚C. Scorpion antivenom from Butantan Institute was used as primary antibody (diluted 1:100 in 1% MPBS). As positive control, a non-immune horse serum was used to coat the plate, while as negative controls, wells were coated with TsV and Ts5, and non-immune horse serum was used as primary antibody (diluted 1:100 in 1% MPBS). As secondary antibody, rabbit anti-horse-HRP (A6917, Sigma-Aldrich, St. Louis, MO, USA) antibodies were used.

B-cell epitope prediction was performed with ABCpred Server tool (http://crdd.osdd.net/raghava/abcpred/) [37], with a window length of 14 residues, a threshold of 0.7, and the Ts5 amino acid sequence already deposited on database (Uniprot ID: P46115) [32].

## Results

### *T. serrulatus* venom and toxins induce spontaneous nociception

The nociceptive response after TsV and toxins injections showed different degrees of pain behavior in mice (Fig 1A). As expected, toxins acting on Nav channels (Ts1 and Ts5) induced significant nociceptive behavior after 10 min of injection (Fig 1B). Ts8, a Kv toxin that modulated Kv4.2 channel, also induced a significant and delayed nociceptive behavior, with the peak of pain behavior observed after 25 min of injection. The other toxins tested (Ts6 and Ts19 frag II), both toxins binding Kv channels, did not showed significant nociceptive behavior.

Notably, TsV and Ts5 presented a significant and lasting (from 5 to 35 min) nociceptive behavior, with a peak of Ts5 at 10 min and 5 min for TsV and with a similar kinetics. Using different quantities of Ts5, the nociceptive behavior demonstrates to be dose-dependent (Fig 1C). Based on the comparable nociceptive kinetic profile of TsV and Ts5, we hypothesized that Ts5 would account for the nociceptive response caused by the whole venom. To test this hypothesis, next we generated anti-Ts5 polyclonal neutralizing antibodies.

### Generation of anti-Ts5 polyclonal antibodies

From 3 immunized animals (Fig 2A), only one of them (named M2) was able to produce detectable titres of anti-Ts5 antibodies (Fig 2B). The i.m. injection of Ts5 showed to be the best route to the neurotoxin inoculation, since we did not have any anti-Ts5 antibody detection when we used intraperitoneal route. The serum of the M2 mouse was used to isolate specific anti-Ts5 IgGs. Mouse IgGs were purified properly, as can be observed in the SDS-PAGE gel (Fig 3A). After purification, the IgGs' showed to keep the Ts5 recognition, although in a lower title (Fig 3B).

### Nociception response induced by Ts5-neutralized venom

Through the nociception assay after incubation of TsV with the producing anti-Ts5, a significant decrease in the mice nociceptive response was observed, when compared to the animals that received the challenge without pre-incubation with the anti-Ts5 antibody (Fig 4A). Nevertheless, the pain behavior was still significant in this group in comparison to the control, showing that other toxins that compose the venom cocktail also contribute to the nociceptive effect (Fig 4B).

Regarding to control group that received isolated Ts5 toxin pre-incubated with the anti-Ts5 antibodies, the nociceptive response was completely abolish, indicating that the antibodies produced by the immunized M2 mouse were able to recognize Ts5 and neutralize its induced nociception.

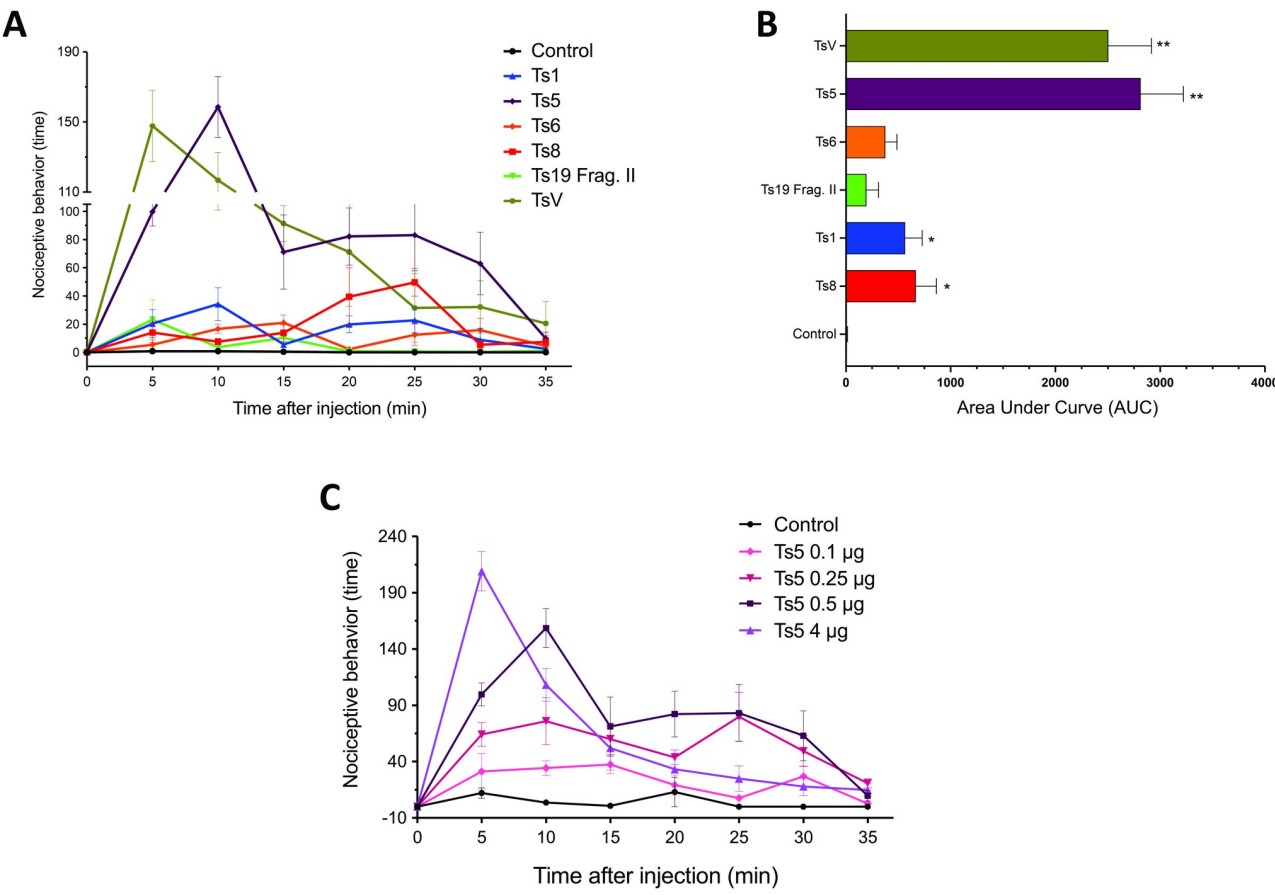

**Fig 1. Nociceptive assays. (A)** Nociceptive behavior was evaluated in C57BL/6 male mice (n = 6) using 0.5 μg paw injections of TsV and isolated toxins (Ts1, Ts5, Ts8, Ts19 frag II). Animals were observed for 35 min to record the time the animals spent either licking or lifting/shaking the injected paw. **(B)** Area under curve (AUC) over any individual parameter in the evaluation of nociceptive response. Data was analyzed using one-way ANOVA and Tukey's post hoc test. * $p < 0.05$ and ** $p < 0.001$, when compared to the control group. **(C)** Mice nociceptive behavior following paw injection with different quantities of Ts5 (0.1–4 μg). Graphs were generated using GraphPad Prism Version 9.3.1.

## Ts5: Immunorecognition and prediction of B-cell epitopes

From indirect ELISA assay, it was demonstrated that scorpion antivenom was able to recognize TsV and Ts5 (Fig 5A), indicating that the specific Brazilian scorpion antivenom is also effective for the treatment of pain in envenoming victims. The *in silico* analysis with Ts5 toxin suggested 4 predicted epitopes (Fig 5B), that could be responsible to antibody binding (Fig 5C).

## Discussion

SSE cause diverse and complex clinical manifestations ranging from local effect to systemic inflammatory reaction, similar to those associated with systemic inflammatory response syndrome and acute sepsis [41]. Scorpion venom is composed of a mixture of molecules, including ion channel-modulating toxins (*i.e.*, neurotoxins) [5, 16, 20]. Neurotoxins can act on sodium, potassium, calcium, and chloride channels being able to induce a high flux of neurotransmitters that is considered the major driver of the pathology following a scorpion sting [6]. Moreover, many of the toxins that affect ion channels are also able to activate the immune system inducing exacerbated inflammatory responses, which is a research area that has been recently explored [13, 14, 20, 42].

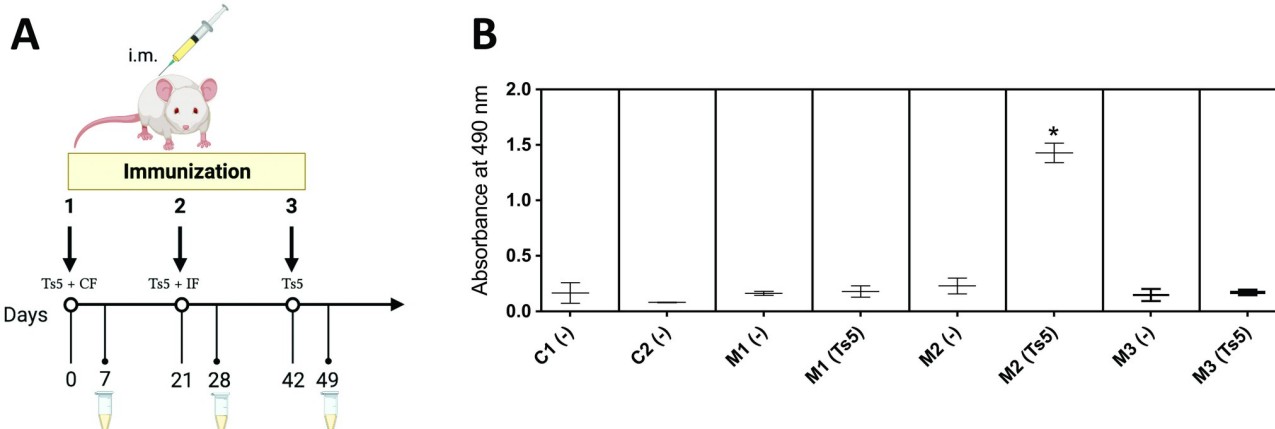

**Fig 2. Ts5-immunization assay. (A)** Male BALB/c mice (n = 3) received intramuscularly (i.m.) 7.5 μg of Ts5 antigen emulsified in a solution containing complete Freunds' adjuvant (CF), followed by two sequential i.m. boosts with 3-week intervals of 7.5 μg and 5 μg/animal, respectively. The second boost with Ts5 was diluted in a solution with incomplete Freunds' adjuvant (IF), and the third boost consisted only by the antigen (Ts5). Control animals (n = 2) received solely saline solution (vehicle). Figure was created with BioRender.com (Agreement number: BO23GVKOCK). **(B)** Immunorecognition of Ts5 toxin (2 μg/well) by animal's sera used in experimental immunization. C (-): mice sera without immunization; M1(-): Mouse 1 serum before immunization; M1: Mouse 1 immunized with Ts5; M2 (-): Mouse 2 serum before immunization; M2: Mouse 2 immunized with Ts5; M3(-): Mouse 3 serum before immunization; M3: Mouse 3 immunized with Ts5. The ELISA assay was performed in duplicate, and results are plotted as mean ± SD. Data was analyzed using one-way ANOVA and Tukey's post hoc test. Graph was generated using GraphPad Prism Version 9.3.1. *$p < 0.05$, when compared to negative controls.

Immediately after a sting caused by Ts scorpion, local pain is the primary manifestation and, although further studies are needed to lighten this event, cutaneous hyperalgesia and local swelling result in a quick local inflammation with the production of pro-inflammatory mediators, amplifying the pain intensity [14]. For instance, Ts5 toxin acts as a pro-inflammatory mediator through its ability to stimulate the release of macrophage TNF-**α** and IL-6 [16]. Moreover, TsV have showed the ability to affect all biological systems such as nervous, cardiac,

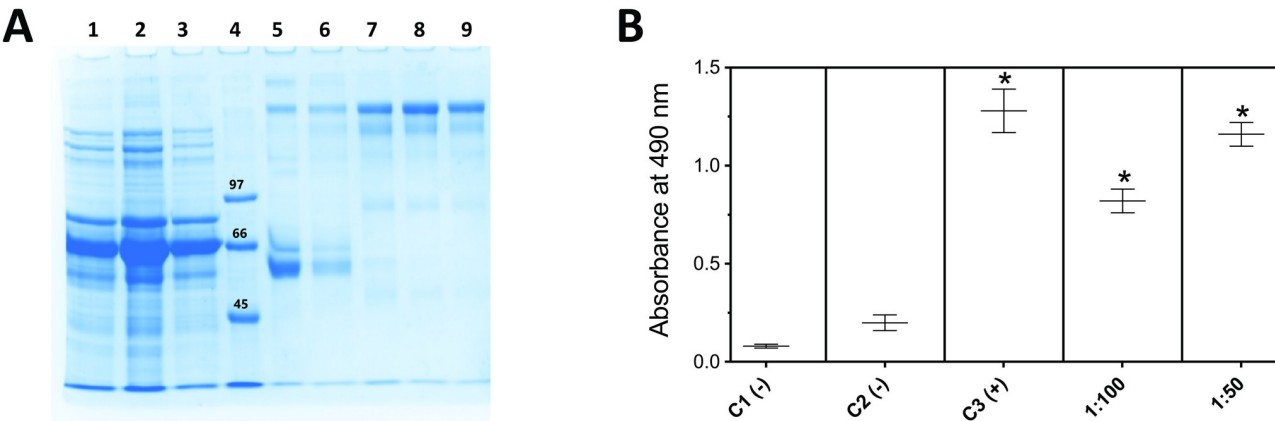

**Fig 3. Purification of IgG anti-Ts5 antibodies. (A)** SDS-PAGE (8%) under reducing conditions following M2 mice-derived serum IgG isolation. Lane 1: mice serum (M2); Lane 2: void; Lane 3: washing; Lane 4: Molecular mass marker (97.0–14.4 kDa); Lane 5: washing 2; Lane 6: washing 3; Lane 7: Proteins (IgGs) eluted on tube 1; Lane 8: Proteins (IgGs) eluted on tube 2; 9: Proteins (IgGs) eluted on tube 3. SDS-PAGE gel was stained with Coomassie brilliant blue. **(B)** Immunorecognition of Ts5 toxin by the pooled anti-Ts5 IgGs' eluted (tubes 1–3). C1(-): wells non sensitized; C2(-): wells sensitized with Ts5 (2 μg) and tested with non-immune mouse serum; C3(+): wells sensitized with mouse non-immune serum; 1:100 and 1:50: wells sensitized with Ts5 (2 μg) and tested with purified mouse anti-Ts5 IgGs. Rat anti-mouse-HRP antibodies were used in the ELISA detection. The ELISA assay was performed in duplicate, and results are plotted as mean ± SD. Data was analyzed using one-way ANOVA and Tukey's post hoc test. Graph was generated using GraphPad Prism Version 9.3.1. *$p < 0.05$, when compared to negative controls.

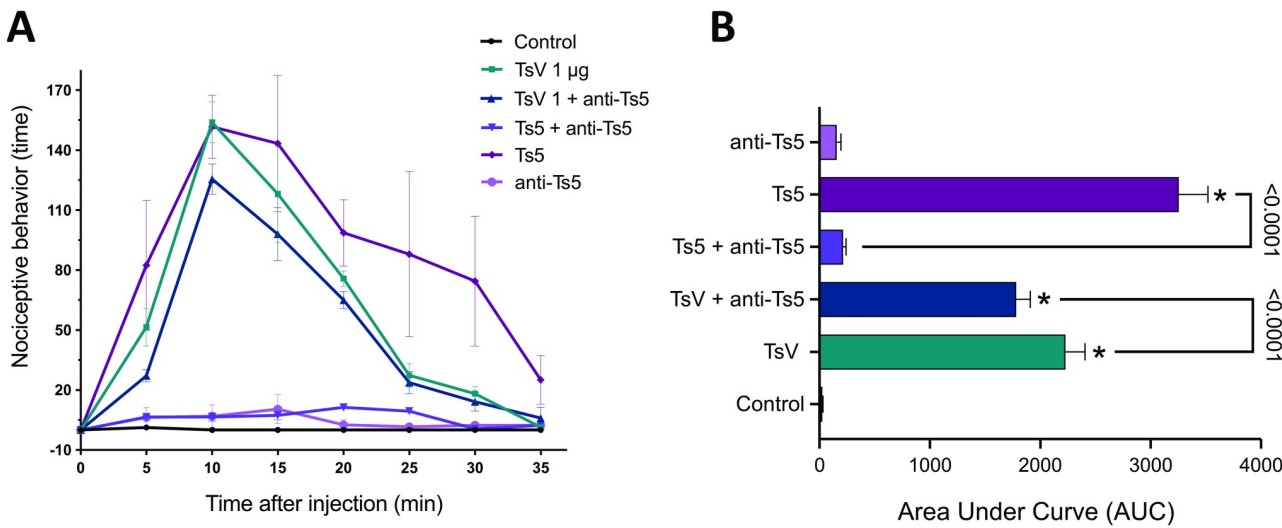

**Fig 4. Nociception assays with Ts5 response neutralized. (A)** Nociceptive behavior was evaluated in BALB/c male mice (n = 6) using 0.5 μg/animal paw of TsV as positive control animals. In the tested group of animals, TsV (1 μg/animal paw) was pre-incubated 1 h at 37°C with 2 eqM of purified IgGs anti-Ts5 (considering the % of Ts5 from the whole venom). The same rate was used in the positive control group, using 0.5 μg of Ts5/animal paw pre-incubated 1 h at 37°C with 2 eqM of IgGs anti-Ts5. Animals were observed for 35 min to record the time the animals spent either licking or lifting/shaking the injected paw. **(B)** Area under curve (AUC) over any individual parameter in the evaluation of nociceptive response with Ts5 neutralized. Data was analyzed using one-way ANOVA and Tukey's post hoc test. * $p < 0.05$, when compared to the control group. Graphs were generated using GraphPad Prism Version 9.3.1.

pulmonary, urinary, digestive, reproductive, muscular, and the immune system [5], which makes TsV a potent lethal cocktail. In this work we performed assays to elucidate how TsV and its different classes of neurotoxins can act on nociceptive response.

Nociception can be defined as the detection of actual or threatened tissue injury; the specialized neural apparatus fulfilling this role is highly evolutionary conserved, providing a key protective role to the organism [43]. Nociceptors are specialized primary sensory neurons innervating target tissue, such as skin and muscle, which detect injurious stimuli (extremes of temperature, high pressure, endogenous mediators, scorpion stings, among others) [44]. Each of these neurons expresses unique repertoires of voltage-gated sodium channels which are key determinants of excitability integrating the propagation of action potentials to the central nervous system and neurotransmitter release [45], and the activation of the nociceptive system will give rise to the perception of pain.

This work pioneering investigated the nociception of different classes of Ts neurotoxins (Ts1: β-NaTx; Ts5: α-NaTx; Ts6: α-KTx; Ts8 and Ts19: β-KTx). The β-NaTx (such as Ts1), similarly to α-NaTx counterparts, can compete for binding sites on both insect and mammalian Nav channels. Indeed, the α-NaTx Ts5 demonstrated to bind to both mammalian and insect channels [16, 46]. In our work, Ts5 was able to promote intense and dose-dependent spontaneous nociception, with a similar profile triggered by TsV, differently from the other tested toxins (Ts1, Ts6, Ts8, and Ts19 frag II). Among the toxins tested, Ts8 is the only one with a role on Kv4.2 being able to induce pain [23]. However, the experimental pain behavior induced by Kv4.2 showed to be mild and prolonged, different from the pain observed after Ts5 administration.

In a former study, authors speculated that TsV could promote a marked ipsilateral nociceptive response, characterized by thermal and mechanical allodynia and paw licking behaviour. Interestingly, that nociception was inhibited by intraperitoneal injection of anti-inflammatory drugs such as indomethacin, dipyrone, and cyproheptadine, or potent analgesic as morphine,

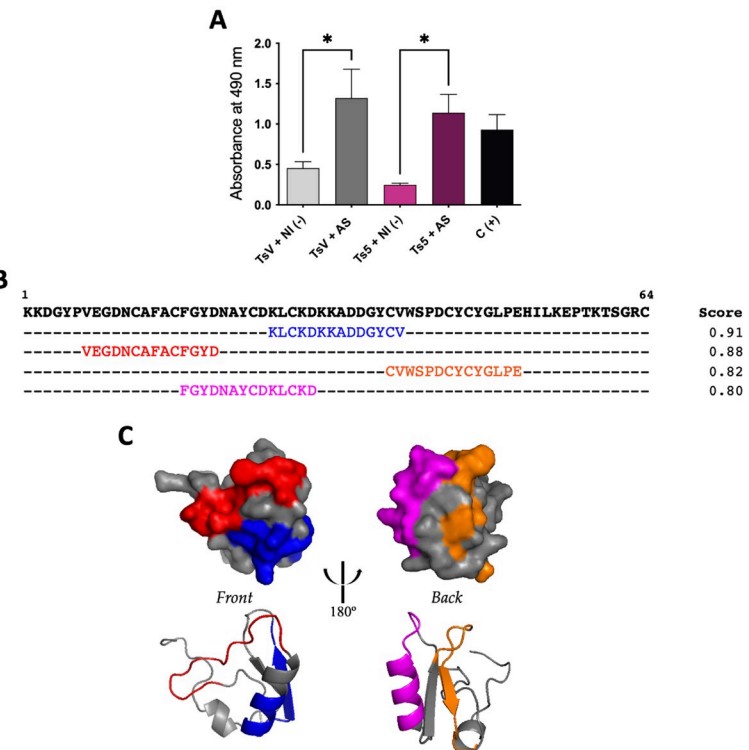

**Fig 5. Ts5 recognition by scorpion antivenom and predicted epitopes. (A)** TsV + NI (-): TsV + non-immune horse serum; TsV + AS: TsV + scorpion antivenom (Butantan Institute); Ts5 + NI (-): Ts5 + non-immune horse serum; Ts5 + AS: Ts5 + scorpion antivenom (Butantan Institute); C (+): wells coated with non-immune horse serum. Rabbit anti-horse-HRP antibodies were used for the ELISA detection. The ELISA assay was performed in duplicate and results are plotted as mean ± SD. Data was analyzed using one-way ANOVA and Tukey's post hoc test. Graph was generated using GraphPad Prism Version 9.3.1. $^{*}p < 0.05$, when compared to the control group. **(B)** Ts5 primary sequence (P46115) [32] with the B-cell epitopes predicted by ABCpred Server tool highlighted and their scores. **(C)** Ts5 molecular model from AlphaFold Protein Structure Database [38, 39], modified by PyMol tool [40], with front and back view. Colors (blue, red, orange, and magenta) are not related to the alpha-helix and beta-pleated sheets, they represent the four B-cell epitopes predicted in different types of 3D representation: surface (up) and ribbon (down).

which leads the authors to conclude that the nociceptive response could result from the action of multiple mediators including eicosanoids, histamine, and 5-hydroxytryptamin [11]. We do not discard that hypothesis; however, based on our *in vivo* assays along with electrophysiological studies, Ts5 activates a nociceptive response immediately after Ts5 injection, inducing an intense pain behaviour in animals (see S1 Video), which could not be a result of mediators of inflammation since they will need time to be produced. Actually, after the stimulus with TsV, macrophages up-regulate the expression of specific receptors, which promote the activation of transcription factors targeting the genes that code for inflammatory mediators leading to a time-consuming inflammatory response [47].

Indeed, using two-electrode voltage-clamp analysis, Ts5 showed to directly inhibit the rapid inactivation of voltage-gated sodium channels related to pain such as Nav 1.3, Nav 1.6, and Nav 1.7 [16]. Such Ts5 activity on a variety of ion channels also explains its high toxicity in mice, although it presents a higher intravenous $LD_{50}$ higher than the major TsV-derived toxin, named Ts1 (76 ± 9 μg/kg for Ts1 *vs*. 94 ± 7 μg/kg for Ts5) [48, 49]. Thus, we can infer that among the explored neurotoxins from TsV, Ts5 is the one triggering the most painful response and Ts1 the highest toxicity.

To verify if the TsV painful stimuli was a result mainly derived from Ts5 effect, we produced anti-Ts5 polyclonal antibodies. Only one of three immunized mice was able to produce antibodies specific to Ts5. Also, as the route of administration dictates the antibody production [50], the i.m. route showed to be important for the production of antibodies anti-Ts5. Since Ts5 presents only 4 predicted epitopes (minimal peptide epitopes), the toxin injection through i.m. route would force antigen presentation to T cells by tissue resident dendritic cells, preventing the induction of T cell tolerance or anergy [51]. In fact, the scorpion antivenom used in Brazil is produced in horses immunized intramuscularly, and we demonstrated here that the horse-derived scorpion antivenom can recognize Ts5.

Analysis of the TSV nociceptive behaviour using anti-Ts5 antibodies, showed that the neutralization of Ts5 from the whole venom was able to significantly reduce the nociception in mice. On the other and, even overturning the Ts5 effect from TsV, the painful stimuli are still significant, demonstrating that other toxins are also contributing to the scorpion painful sting. Therefore, the intense pain sensitivity induced by Ts sting is a result of the binding of different toxins on a gamut of receptors and the induced production of inflammatory mediators.

## Conclusion

Our study explored the nociception behavior induced by different classes of scorpion neurotoxins (α-NaTx, β-NaTx, α-KTx, and β-KTx) and demonstrated the important role of Ts5 on Ts painful sting. Thus, in addition to providing new insights to understand the intense pain associated with Ts envenoming, this study could also contribute for the development of novel and specific therapies.

## Supporting information

**S1 Video. Nociception video.**
(WMV)

## Author Contributions

**Conceptualization:** Manuela Pucca.

**Methodology:** Felipe Cerni, Isadora Oliveira, Francielle Cordeiro, Karla Bordon.

**Project administration:** Wuelton Monteiro, Eliane Arantes, Thiago Cunha, Manuela Pucca.

**Supervision:** Eliane Arantes, Manuela Pucca.

**Writing – original draft:** Isadora Oliveira, Francielle Cordeiro, Karla Bordon, Isabela Ferreira, Wuelton Monteiro, Manuela Pucca.

**Writing – review & editing:** Felipe Cerni, Isadora Oliveira, Francielle Cordeiro, Karla Bordon, Isabela Ferreira, Wuelton Monteiro, Eliane Arantes, Thiago Cunha, Manuela Pucca.

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
