## [Decision Letter · Decision Letter 0]

21 Sep 2022

Dear Dr Pucca,

Thank you for submitting your manuscript "The nociceptive response induced by different classes of Tityus serrulatus neurotoxins: the important role of Ts5 in venom-induced nociception" (PNTD-D-22-01074) for review by PLOS Neglected Tropical Diseases. I have received two reviews of your paper, all of which indicate that some major revision is required before it can be considered suitable for publication in our journal. Therefore, if you feel that you can suitably address the comments (included below), I invite you to revise and resubmit your manuscript.

When resubmitting your manuscript, please carefully consider all issues mentioned in the reviewers' comments, make the requested changes in the manuscript and indicate their place in the text (e.g. line numbers), outline every change made point by point, and provide suitable rebuttals for any comments not addressed.

We cannot make any decision about publication until we have seen the revised manuscript and your response to the reviewers' comments. Your revised manuscript is also likely to be sent to reviewers for further evaluation.

Sincerely,

Philippe BILLIALD

Academic Editor

José María Gutiérrez

Section Editor

Dear Authors,

Thank you for submitting your manuscript "The nociceptive response induced by different classes of Tityus serrulatus neurotoxins: the important role of Ts5 in venom-induced nociception" (PNTD-D-22-01074) for review by PLOS Neglected Tropical Diseases. I have received two reviews of your paper, all of which indicate that some major revision is required before it can be considered suitable for publication in our journal. Therefore, if you feel that you can suitably address the comments (included below), I invite you to revise and resubmit your manuscript.

When resubmitting your manuscript, please carefully consider all issues mentioned in the reviewers' comments, make the requested changes in the manuscript and indicate their place in the text (e.g. line numbers), outline every change made point by point, and provide suitable rebuttals for any comments not addressed.

Reviewer's Responses to Questions

**Key Review Criteria Required for Acceptance?**

**Methods**

-Are the objectives of the study clearly articulated with a clear testable hypothesis stated?

-Is the study design appropriate to address the stated objectives?

-Is the population clearly described and appropriate for the hypothesis being tested?

-Is the sample size sufficient to ensure adequate power to address the hypothesis being tested?

-Were correct statistical analysis used to support conclusions?

-Are there concerns about ethical or regulatory requirements being met?

Reviewer #1: See comments to authors

Reviewer #2: Needs improvement, comments below.

**Results**

-Does the analysis presented match the analysis plan?

-Are the results clearly and completely presented?

-Are the figures (Tables, Images) of sufficient quality for clarity?

Reviewer #1: Specific points that need the authors' attention:

1.- Spontaneous nociception?? As pain cannot be directly measured in rodents, many methods that quantify “pain-like” behaviors or nociception have been developed. These behavioral methods can be divided into stimulus-evoked or non-stimulus evoked (spontaneous) nociception, based on whether or not application of an external stimulus is used to elicit a withdrawal response. Authors induced a pain response by injecting Ts venom or isolated Ts neurotoxins. As far as I understand this action should be regarded as stimulus-evoked nociception rather than spontaneous nociception... 

2.- Toxin concentrations were calculated using ProtParam, an algorith accessible in the Expasy server. Theoretical extinction coefficients calculated from the amino acid sequence of a protein were developed before the turn of the (XXI) century and are poorly known by most researchers of the omics era. It is thus relevant to cite original reference(s) and briefly describe, in the corresponding manuscript's section, the methodology underlying the calculation. 

3.- The statement "Knowing that IgGs are bivalents, hypothetically, the ratio of binding site and antigen would be 4:1" is cryptic (to me). Do you mean that IgG bears 4 antigen-binding sites per molecule? Or that the molar ratio of IgG:antigen is 4:1? Please, revise and/or clarify! 

4.- "The same rate was used in the positive control group, using 0.5 ug of Ts5/animal paw pre-incubated 1 h at 37° C with 2 eqM of IgGs anti-Ts5". Please, specify the method used to quantify IgG. 0.5 ug Ts5 = 69.5 pmoles; 2eqM IgG = (2 x 69.5 x 10E-12) moles x 150000 g/mol = 20.85 ug. Is this correct? 

5.- Table 2. Please clarify the meaning of the two columns for "ug" (mass and e). If I undertood it correctly the first column should be Absobance units and the second column (A/e = concentration of toxin in mg/mL). However, the fact that Figure 4 refers to using 0.5 ug in the nociception assays induce to confusion! Similarly, in M&M: "For spontaneous nociceptive test, animals (n = 6) were challenged with intraplantar injection of TsV 95 or its isolated toxins (Ts1, Ts5, Ts6, Ts8, and Ts19 frag II) using 0.5 μg/paw (right hind paw)". Third column correspond to umoles of toxins which when dissolved in 10 uL give the listed molarity (uM). These figures are nowhere used in the manuscript; why are they listed in Table 2; All this should be better explained in the text and the columns headers should be labelled accordingly. 

6.- "The in silico analysis with Ts5 toxin demonstrated 4 predicted epitopes (Fig. 5B), that could be responsible to antibody binding (Fig. 5C)". Demonstrated? You may mean "suggested". The server is able to predict continuous B-cell epitopes with 65.93% accuracy using recurrent neural network (Saha, S and Raghava G.P.S. (2006) Prediction of Continuous B-cell Epitopes in an Antigen Using Recurrent Neural Network. Proteins,65(1),40-48). On the other hand, The suggested four epitopes cover almost the whole sequence of Ts5. The conclusion from the in silico analysis should be that the linear sequence of Ts5 may harbor one or more B-cell epitopes. Any evidence that any of the 4 predicted epitopes is actually recognized by the anti-Ts5 antibodies?, i.e. can synthetic peptides mimicking the predicted epitopes compete with anti-Ts5 antibodies for Ts5 binding?

Reviewer #2: Needs improvement, comments below.

**Conclusions**

-Are the conclusions supported by the data presented?

-Are the limitations of analysis clearly described?

-Do the authors discuss how these data can be helpful to advance our understanding of the topic under study?

-Is public health relevance addressed?

Reviewer #1: See comments to authors

Reviewer #2: Needs improvement, comments below.

**Editorial and Data Presentation Modifications?**

Reviewer #1: (No Response)

Reviewer #2: (No Response)

**Summary and General Comments**

Reviewer #1: This is an interesting paper that addresses a poorly studied/neglected aspect of the Brazilian yellow scorpion, Tityus serrulatus (Ts), stings: the molecular basis underlying nociceptive response induced by whole Ts venom as well as by its isolated neurotoxic components. In addition, Cerni and colleagues developed a mice-derived antibody targeting Ts5 and investigated its capability to abolish nociception produced by this toxin in the context of whole Ts venom.

Reviewer #2: The article addresses a relevant topic, given the reduced number of studies that effectively assess the role of isolated Tityus toxins in the context of nociception. 

However, some conclusions are hasty based on the results found.

- Evaluate the choice of concentrations used to prove the effects of Ts5, considering the percentage of this toxin in the venom, as well as the total amount of venom injected in an accident.

- The use of polyclonal antibodies does not seem appropriate to evaluate the participation of a single toxin (Ts5) in the context of nociception, considering the percentage of identity among different toxins present in T. serrulatus venom. 

- The results obtained from in silico analysis of epitope prediction do not contribute to the discussion of the results, since no validation with the indicated epitopes was performed (e.g. isolation by immunoaffinity of specific polyclonal antibodies and their characterization).

-The identification of exclusive epitopes and their use as immunogens for the production of protective antibodies could validate the importance of the neutralization of this toxin, in the context of the venom.

-The use of specific monoclonal antibodies to Ts5 could also be useful to inhibit the activities of this toxin.

-The evaluation of neutralization should have been performed with different concentrations of antibodies (confirm dose dependency).

- The literature cited deserves an update and citations need to be adjusted ( eg reference 38)

- The video alone, without the proper controls, does not provide any conclusions.

PLOS authors have the option to publish the peer review history of their article (what does this mean?). If published, this will include your full peer review and any attached files.

Reviewer #1: Yes: Juan J. Calvete

Reviewer #2: No
---

## [Decision Letter · Decision Letter 1]

23 Dec 2022

Dear Dr Pucca,

We are pleased to inform you that your manuscript 'The nociceptive response induced by different classes of Tityus serrulatus neurotoxins: the important role of Ts5 in venom-induced nociception' has been provisionally accepted for publication in PLOS Neglected Tropical Diseases.

Best regards,

Philippe BILLIALD

Academic Editor

José María Gutiérrez

Section Editor

Reviewer's Responses to Questions

**Key Review Criteria Required for Acceptance?**

**Methods**

-Are the objectives of the study clearly articulated with a clear testable hypothesis stated?

-Is the study design appropriate to address the stated objectives?

-Is the population clearly described and appropriate for the hypothesis being tested?

-Is the sample size sufficient to ensure adequate power to address the hypothesis being tested?

-Were correct statistical analysis used to support conclusions?

-Are there concerns about ethical or regulatory requirements being met?

Reviewer #1: YES to all, except the las point: "There are NO concerns about ethical or regulatory requirements being met"

Reviewer #2: (No Response)

**Results**

-Does the analysis presented match the analysis plan?

-Are the results clearly and completely presented?

-Are the figures (Tables, Images) of sufficient quality for clarity?

Reviewer #1: YES

Reviewer #2: (No Response)

**Conclusions**

-Are the conclusions supported by the data presented?

-Are the limitations of analysis clearly described?

-Do the authors discuss how these data can be helpful to advance our understanding of the topic under study?

-Is public health relevance addressed?

Reviewer #1: YES

Reviewer #2: (No Response)

**Editorial and Data Presentation Modifications?**

Reviewer #1: (No Response)

Reviewer #2: (No Response)

**Summary and General Comments**

Reviewer #1: (No Response)

Reviewer #2: (No Response)

PLOS authors have the option to publish the peer review history of their article (what does this mean?). If published, this will include your full peer review and any attached files.

Reviewer #1: **Yes: **Juan J. Calvete

Reviewer #2: No

---

## [Editor Report · Acceptance letter]

16 Jan 2023

Dear Dr. Pucca,

We are delighted to inform you that your manuscript, "The nociceptive response induced by different classes of Tityus serrulatus neurotoxins: the important role of Ts5 in venom-induced nociception," has been formally accepted for publication in PLOS Neglected Tropical Diseases.

Best regards,

Shaden Kamhawi

co-Editor-in-Chief

Paul Brindley

co-Editor-in-Chief
